



# Coda-derived source properties estimated using local earthquakes in the Sea of Marmara, Türkiye

Berkan Özkan[1], Tuna Eken[1], Peter Gaebler[2], Tuncay Taymaz[1]

[1] Department of Geophysical Engineering, Faculty of Mines, Istanbul Technical University, Maslak, Istanbul, Türkiye
[2] BGR Federal Institute for Geosciences and Natural Resources, Stilleweg 2, 30655, Hannover, Germany

*Correspondence to*: Berkan Özkan (ozkanber@itu.edu.tr)

**Abstract.** Accurate estimates of the moment magnitude of earthquakes that physically measures the earthquake source energy are crucial for improving our understanding of seismic hazards in regions prone to tectonic activity. To address this need, a method involving coda wave modelling was employed to estimate the moment magnitudes of earthquakes in the Sea of

Marmara. This approach enabled to model the source displacement spectrum of 303 local earthquakes recorded at 49 seismic stations between 2018 and 2020 in this region. The coda wave traces of individual events were inverted across twelve frequency ranges between 0.3 and 16 Hz. The resultant coda-derived moment magnitudes were found to be in good accordance with the standard local magnitude estimates. However, the notable move-out between local magnitude and coda-derived moment magnitude estimates for smaller earthquakes less than a magnitude of 3.5 likely occurs due to potential biases arising from

incorrect assumptions for anelastic attenuation and the finite sampling intervals of seismic recordings. Scaling relations between the total radiated energy and seismic moment imply a nonself-similar behaviour for the earthquakes in the Sea of Marmara. Our findings suggest that larger earthquakes in the Sea of Marmara exhibit distinct rupture dynamics compared to smaller ones, resulting in a more efficient release of seismic energy. In conclusion, here we introduce an empirical relationship devised from the scatter between local magnitude and coda-derived moment magnitude estimates.

**1 Introduction**

Having a strong and consistent understanding of source properties, such moment magnitude estimates, is extremely important in tectonically active region such as the Sea of Marmara located at the northwest of the North Anatolian Fault Zone in Türkiye. This is essential for accurately assessing seismic hazard potential, as it primarily relies on creating dependable seismicity catalogues. Besides, precise data on source parameters plays a significant role in the development of regional attenuation

properties.

Traditional magnitude scales such local, body wave, or surface wave magnitude scales ($M_L$, $m_b$, $M_S$) derived from direct wave analyses may exhibit bias due to various factors including source radiation pattern, directivity, and path heterogeneities. These effects can cause significant changes in direct wave amplitude measurements (e.g., Favreau and Archuleta, 2003). Over the past four decades since Aki's work in 1969, computational seismology has achieved remarkable progress, enabling the



integration of scattered wavefields, i.e., coda waves, into studies of source parameters (e.g., Sato et al., 2012). These developments have expanded our understanding of seismic events and improved the accuracy of source parameter estimation. Aki and Chouet, (1975) observed that these scattered wave train and its spectral content behave similarly at the recordings of different stations for a given earthquake. They further noticed coda duration is independent from the azimuth or epicentral distance. More recently, studies analysing local and/or regional coda envelopes suggest that coda wave amplitudes are notably

less variable, about 3 to 5 times, compared to direct wave amplitudes (e.g., Mayeda and Walter, 1996; Mayeda et al., 2003; Eken et al., 2004; Malagnini et al., 2004; Gök et al., 2016). It is widely recognized that local or regional coda waves mainly consist of scattered waves. These wave trains can be explained by Aki's single-scattering model (1969), which is significantly less sensitive to source radiation pattern effects compared to direct waves, owing to the volume-averaging property of coda waves that sample the entire focal sphere (e.g., Aki and Chouet, 1975; Rautian and Khalturin, 1978). For a more in-depth

understanding of coda generation theory and advances in empirical observations and modelling efforts, refer to Sato et al. (2012).

Various methods depending on coda waves analysis have been utilized for earthquake source scaling. They are usually categorized into two groups. The first group of methods is known as the parametric approach and involves employing a coda normalization strategy. This requires applying corrections (including path effect, S-to-coda transfer function, site effect, and

any distance-dependent changes in coda envelope shape) on the measurements extracted from coda wave envelopes through empirically derived quality factors that account for seismic attenuation parameters (e.g., intrinsic and scattering factors) or site effect caused by near surface geology conditions. To determine the final source properties, reference events with pre-estimated seismic moments based on waveform inversion techniques are used. Forward calculation of the synthetic coda envelopes is achieved by using either single-backscattering or more advanced multiple-backscattering approximations (Sato et al., 2012).

Empirical coda envelope methods have been successfully applied in regions with complex tectonics, such as northern Italy (e.g., Morasca et al., 2008), Türkiye and the Middle East (e.g., Mayeda et al., 2003; Eken et al., 2004; Gök et al., 2016), and the Korean Peninsula (e.g., Yoo et al., 2011).

The approaches in the second group involve estimating source and structural properties using a joint inversion technique in which source-, path-, and site-specific factors are optimized simultaneously by comparing the observed coda envelope with its

physically derived representative synthetic coda envelope within a selected time window including both the observed coda and direct S-wave parts. While the conventional coda normalization method corrects for undesired effects of source and site amplifications, it may not work well for small events with short coda lengths. This occurs mainly due to dominating random seismic noise that disrupts the requirement of a homogeneous coda wave energy distribution in space. To overcome this limitation, we incorporate source excitation and site amplification terms in the inversion process in which synthetic coda wave

envelopes are analytically expressed via the radiative transfer theory (RTT). The RTT was originally implemented on coda waves by Sens-Schönfelder and Wegler (2006), and has been successfully tested on local and regional earthquakes ($4 \leq M_L \leq 6$) detected by the German Regional Seismic Network. Moreover, it has been applied to investigate source- and frequency-dependent attenuation properties in various geological settings, including the upper Rhine Graben and Molasse basin regions



in Germany, western Bohemia–Vogtland in Czechia (Eulenfeld and Wegler, 2016), the entire United States (Eulenfeld and
Wegler, 2016), and the central and western North Anatolian Fault Zone (Gaebler et al., 2019; Izgi et al., 2020). Previous studies
(Gusev and Abubakirov, 1996) have further considered a more realistic earth model with anisotropic scattering conditions,
resulting in peak broadening effects of direct seismic wave arrivals. The propagation of P-wave elastic energy and the
conversion between P- and S-wave energies with this approach has been used in Zeng et al. (1991), Przybilla and Korn (2008),
and Gaebler et al. (2015a).

In this study, we generate source spectra for 303 local events with magnitudes $2.5 \leq M_L \leq 5.7$ that occurred in Marmara Sea
region as the product of a joint inversion of S-wave and coda wave components extracted. To estimate coda-derived source
spectra and further moment magnitude and total radiated seismic energy of these selected earthquakes we utilized an open-
source python based Qopen software (Eulenfeld, 2020), which employs the isotropic acoustic radiative transfer theory (RTT)
to calculate synthetic coda envelopes. Gaebler et al. (2015a) have noted that modelling outcomes from isotropic scattering
were nearly equivalent to those inferred from more complex elastic RTT simulations with anisotropic scattering conditions.
Adopting the joint inversion technique offers advantages, as it remains unaffected by potential biases that could arise from
external information, such as i.e., source properties of a reference earthquake that are separately estimated and then used for
calibration in coda-normalization methods. The advantage of the approach exploited in this work stems from the analytical
expression of a physical model incorporating source- and path-related parameters to describe the scattering process.
Furthermore, the optimization process during the joint inversion enables source parameter estimates for relatively small-sized
events compared to those employed in coda normalization methods.

## 2 Regional Settings and Seismic Hazard Potential

Our study area is the Sea of Marmara, located in the northwest of the 1600-km-long North Anatolian Fault Zone (NAFZ). This
fault zone is an intercontinental dextral strike-slip fault that represents as a boundary between the Eurasian plate to the north
and the Anatolian plate to the south. The tectonic activity in this region is primarily the result of the collision between the
Arabian and Eurasian plates to the east and southwest-trending rollback of the Hellenic subduction zone in the south Aegean
Sea to the west (e.g., McClusky et al., 2000; McKenzie, 1972).

The NAFZ has experienced numerous devastating historical earthquakes that have ruptured throughout its the entire length
with an overall westward migrating pattern (Stein et al., 1997). The first major earthquake of significant consequence within
our specific area of interest occurred along the Ganos segment situated at the westernmost part of the NAFZ in 1912. More
recently, two destructive earthquakes, namely the Izmit earthquake ($M_w$ 7.4, August 17, 1999) and the Düzce earthquake ($M_w$
7.2, November 12, 1999), have affected the northwestern branch of the NAFZ. A study by Barka et al. (2002) on the historical
earthquake records reported in Ambraseys and Jackson (2000) has revealed the region lying between the 1912 and 1999
ruptures represents a seismic gap in the Sea of Marmara.



The NAFZ divides into shorter segments and becomes discontinuous as it extends westward, (e.g., Barka and Kadinsky-Cade, 1988). Within a marine basin about 280-km-long and 80 km wide, the fault crosses the Sea of Marmara. It is characterized by various complex structures that have been formed due to the interaction between extensional and strike-slip shear deformation processes (Gürer et al., 2006, Taymaz et al., 2004; Taymaz et al., 2007). Beneath the Sea of Marmara, the fault is divided into three segments. The first one is the 15 km long Ganos segment, which might have experienced rupture during the 1912

earthquake (e.g., Ambraseys and Finkel, 1987). The second segment is the Central Marmara Segment, stretching 105 km, and has been considered a seismic gap since 1766 (e.g., Okay et al., 2000). Most recently, on 26th September 2019, the Silivri High-Kumburgaz Basin (central Marmara Sea) experienced an earthquake with a magnitude of 5.7. The earthquake ruptured a thrust fault with a minor strike-slip component at the north of the eastern end of this gap, relatively in the shallow depth range (h=8 km) (Irmak et al., 2021). The third segment, the North Boundary segment, covers 45 km and was likely involved

in the 1894 rupture according to (Ambraseys and Finkel, 1987).

Following the 1999 $M_w$ 7.4 Izmit earthquake, Coulomb stress change calculations performed by King et al. (2001) and Durand et al. (2013) demonstrate that new stress accumulation is focused on this western branch in the Sea of Marmara. In fact, precise locations of microseismicity indicated that the two 1999 earthquakes activated seismicity to the south of Istanbul along the northwest branch of the NAFZ beneath the Sea of Marmara (e.g., Bohnhoff et al., 2013; Sato et al., 2004; Schmittbuhl et al.,

2016; Taymaz et al., 2004) and indicate the depth extent of the NAFZ in the crust. The seismic gap along the northern segment of the NAFZ within the Çınarcık Basin at the eastern shear zone of the Sea of Marmara is well identified by high-resolution observations of microseismicity (e.g., Bohnhoff et al. 2013) and geodetic locking depth estimates (Ergintav et al., 2014). Recently, crustal velocity images from a few seismic tomography experiments (e.g., Bayrakci et al., 2013; Tarancıoğlu et al., 2020; Turunçtur et al., 2023) conducted in the region confirmed profound relatively high and low velocity zones consistent

with the locked or aseismically creeping zones. The existing seismic gap of ~150 km unruptured Main Marmara Fault segment (the combination of North Boundary and Central Marmara segment) of the NAFZ beneath the Sea of Marmara has been subject to several studies mainly involving spatio-temporal microseismicity characteristics (e.g., Bohnhoff et al., 2013; Sato et al., 2004; Schmittbuhl et al., 2016; Wollin et al., 2018). This area is predicted to be the location of a potential major earthquake in the future, according to research by Bohnhoff et al. (2013). Therefore, it is crucial to have accurate estimates of the physical

measures of energy released during small-to-moderate size earthquakes to improve seismic hazard assessments in this tectonically active region.

Using Coulomb stress change calculations after the 1999 $M_w$ 7.4 Izmit earthquake King et al. (2001) and later Durand et al. (2013) modelled the new stress accumulation would concentrate on the western branch in the Sea of Marmara. In fact, the precise locations of microseismic activity indicated that the two 1999 earthquakes activated seismicity to the south of Istanbul

along the northwest branch of the NAFZ beneath the Sea of Marmara (e.g., Bohnhoff et al., 2013; Sato et al., 2004; Schmittbuhl et al., 2016; Taymaz et al., 2004). In the eastern shear zone of the Sea of Marmara, the North Boundary segment of the NAFZ, located within the Çınarcık Basin, displays a seismic gap. The presence of this seismic gap has been identified through precise locations of microseismic activity reported in Bohnhoff et al. (2013), and further supported by geodetic locking depth estimates





from Ergintav et al. (2014). Recently, crustal velocity images from seismic tomography experiments (e.g., Tarancıoğlu et al., 2020; Turunçtur et al., 2023) conducted in the region confirmed profound relatively high and low velocity zones consistent with the locked or aseismically creeping zones. These images confirmed the existence of profound relatively high and low velocity zones, consistent with areas that are either locked or aseismically creeping.

The segment of the Main Marmara Fault (a combination of the North Boundary and Central Marmara segments) beneath the Sea of Marmara, spanning approximately 150 km, remains unruptured and represents an existing seismic gap. Numerous studies, particularly focusing on spatio-temporal microseismicity and seismic structure characteristics (e.g., Bohnhoff et al., 2013; Sato et al., 2004; Schmittbuhl et al., 2016; Wollin et al., 2018, Smith et al., 1995; Laigle et al., 2008) have investigated this area extensively. Although primary slip is generally considered to occur on the northern branch of the NAFZ (e.g., Barka, 1992; McClusky et al., 2000; Reilinger et al., 2006) along most of its length as this branch has experienced significant earthquakes with $M_w > 6.9$ during the past century. However, the Marmara segment, located just south of the densely populated city of Istanbul, has not seen major earthquakes (Bohnhoff et al., 2016) as it, thus, is considered a potential location for a major earthquake in the future (Bohnhoff et al., 2013). Bohnhoff et al. (2013) and Ergintav et al. (2014) reported that some of the existing locked segments, i.e., the Princes Islands segment situated directly adjacent to Istanbul, have the potential to generate an earthquake with a magnitude greater than 7. Thus, reliable estimates of the physical measure of the future seismic energy releases of small-to-moderate size earthquakes are of utmost importance for making better seismic hazard assessments in this tectonically active region.

## 3 Data

In this study, we exploited digital waveforms of local earthquake recordings from at 49 broadband seismic stations in the Sea of Marmara between 2018 and 2020 (Fig 1). We benefited from revised earthquake catalogue information acquired from the Kandilli Observatory and Earthquake Research Institute (KOERI) to extract waveform data for a total of 375 examined events with station–event pair distance less than 200 km and focal depths less than 20 km. The majority of seismic activity related to NAFZ in the Sea of Marmara. There are no further requirements, such as taking geographical distribution or azimuthal coverage into account as coda waves provide a path-wide averaging effect (e.g., Mayeda et al., 2003).

At the very beginning we deconvolve the instrument response to better mimic the actual ground motion on seismograms. Our data pre-processing steps involved band-pass filtering of velocity seismograms using a Butterworth type band-pass filter at several frequency bands with central frequencies of 0.3, 0.5, 0.7, 1, 1.4, 2.0, 2.8, 4.0, 6.0, 8.0, 12.0, 16.0 Hz that varied depending on the spectral content of a given event.





Figure 1: Spatial distribution of local events ($2.5 \leq M_L \leq 5.7$) occurred in between 2018 and 2020 are shown with circles color-coded by the focal depths according to the KOERI catalogue. White triangles indicate used stations in the present work.

Later, we performed a Hilbert transform on the filtered waveform data between each frequency bands to generate the total energy envelopes. To predict the P- and S-wave onsets on these envelopes, an average crustal velocity model was employed. Based on this information, several steps taken to ensure to more accurate seismic moment ($M_0$), and thus coda-derived moment magnitude ($M_{w-coda}$) can be given as follows:

    i. The noise level before the P-wave onset was removed,

ii. The S-wave window was defined, starting 8 s prior to and 10 s after the S-wave onset to include all direct S-wave energy effectively,

    iii. Following the S-wave window, a coda window starts at 5 s before and ends 150 s after the S-wave onset or it ends if Signal to Noise Ratio (SNR) of 3.



Here it is worth mentioning that the length of the coda windows might be shortened under two circumstances: when the signal-
to-noise ratio (SNR) is less than 2.5, or when coda waves from two earthquakes (e.g., aftershock sequences) occur within the
same analysis window, leading to an additional rise rather than a decrease in the envelope.

The earthquakes with less than $10\,\mathrm{s}$ of coda length and the earthquakes with the recordings of less than 4 stations were
disregarded by our automated process. We further conducted a visual inspection on each waveform to assure high-quality data.
After applying all these criteria, 6557 station-event pairs from 303 out of 375 all analysed earthquakes ($2.5 \leq M_L \leq 5.7$ within
a radius of 200 km) remained for further data modelling process.

## 4 Method

### 4.1 $M_{w-coda}$ Estimation

We used an inversion scheme adopted by Eken (2019). Procedure was originally developed by Sens-Schönfelder and Wegler
(2006), and later on Eulenfeld and Wegler (2016) modified it to model intrinsic and scattering attenuation parameters.
The forward part dealing with the energy density computation for a particular frequency band assuming a source that emits
radiation uniformly in all directions (isotropic), is given by Sens-Schönfelder and Wegler (2006) as follows,

$$E_{mod}(t,r) = WR(r)G(t,r,g)e^{-bt} \tag{1}$$

where R and W indicate the energy site amplification factor, and source term, respectively. b represents the intrinsic attenuation
parameters. $G(t,r,g)$ indicates the Green's function and considers both direct and scattered wave fields. Its analytical
expression is given by Paasschens (1997) as follow:

$$G(t,r,g_0) = \exp(-v_0 t g_0)\left[\frac{\delta(r-v_0 t)}{4\pi r^2} + \left(\frac{4\pi v_0}{3 g_0}\right)^{-\frac{3}{2}} t^{-\frac{3}{2}}\left(1-\frac{r^2}{v_0^2 t^2}\right)^{\frac{1}{8}} K\left(v_0 t g_0\left(1-\frac{r^2}{v_0^2 t^2}\right)^{\frac{3}{4}}\right) H(v_0 t - r)\right] \tag{2}$$

with $K(x) = e^x\sqrt{1+\frac{2.026}{x}}$

where $g_0$ is the scattering coefficient and $v_0$ is the mean S-wave velocity. In Eq. 2 the term given within the Dirac delta
function describes the direct wave and the rest represents scattered wave part of the Green's function.
Potential differences between predicted and observed energy densities for each earthquake recorded at each station using $N_{ij}$
time samples in a specific frequency band can be minimized by

$$\epsilon(g) = \sum_{i,j,k}^{N_S,N_E,N_{ij}}\left(\ln E_{ijk}^{obs} - \ln E_{ijk}^{mod}(g)\right)^2 \tag{3}$$

where, $N_S$ and $N_E$ represent the numbers of stations (index $i$) and events (index $j$), respectively. Then the scattering attenuation
parameter (g) will be optimized following Eq. 4.





$\quad \ln E_{ijk}^{obs} = \ln E_{ijk}^{mod}(g)$ (4)

Substituting Eq. 1 into Eq. 4 will give Eq. 5

$\ln E_{ijk}^{obs} = \ln G\big(t_{i,j,k}, r_{ij}, g\big) + \ln R_i + \ln W_j + bt_{ijk}$ (5)

Eq. 5 contains $\sum_{i,j} N_{ij}$ equations and $N_S + N_E + 1$ variables as it indicates an overdetermined inversion problem by having $b$, $R_i$, and $W_j$ unknown parameters. Thus Eq. 5 can be solved by using a least-squares approach. $\epsilon(g)$ can be defined by the sum

over the squared residuals of the solution.

Three main steps followed in this inversion scheme to optimize unknown model parameters ($g, b, R_i$, and $W_j$) is given in Eulenfeld and Wegler (2016).

   i.   Calculation of the Green's function for fixed scattering parameters g and minimizing Eq. 5 to solve for $b$, $R_i$, and $W_j$.

ii.  Calculation of $\epsilon(g)$ through Eq. 3.

   iii. Repeating the step i and ii by letting $g$ to vary to find the optimal $b$, $R_i$, and $W_j$, until the error function $\epsilon(g)$ is minimized.

In Fig. 2 we present an example for this minimization process that was applied to the observed coda envelopes at twelve different frequency bands generated by using one selected earthquake recorded at 49 seismic stations of the study area.

The yield of the minimization of the error function $\epsilon(g)$ outlined above will be the spectral source energy term $W_j$, site response $R_i$, and attenuation parameters $b$ and $g$, that satisfy the optimal fitting between observed and predicted coda envelopes.

Using spectral source energy $W$ in frequency domain, source displacement spectrum and thus seismic moment and moment magnitudes can be obtained. (Sato et al., 2012) describe the S-wave source displacement spectrum considering a double-couple source in the far field as,

$\quad \omega M(f) = \sqrt{\dfrac{5\rho_0 v_0^5 W}{2\pi f^2}}$ (6)

Here $W$ is the radiated S-wave energy at a center frequency $f$, $v_0$ is the mean S-wave speed, and $\rho_0$ is the density of the medium.





**Figure 2: Optimization process for the event (30 November 2018 $M_L = 2.9$ and $M_{w-coda} = 3.02$ ) recorded at 23 different stations (frequency band 5.5 Hz - 10.5 Hz). Large panel shows the plot of the $\epsilon$ as a function of $g_0$ for the given frequency band. Blue cross shows the least misfit. Numbered small panels display least square solutions for the different $g_0$ guesses and best fit for optimal $g_0$. Dark grey dots represent the ratio $E_{obs}/G$ and grey lines represents the observed envelopes from different stations. Thick black line is the line to estimate $b$ and $W$ by using its slope.**

Abercrombie (1995) elucidated the correlation between the obtained source displacement spectrum and the seismic moment magnitude by

$$\omega M(f) = M_0 \left(1 + \left(\frac{f}{f_c}\right)^{\gamma n}\right)^{-\frac{1}{\gamma}} \tag{7}$$

where $n$ and $\gamma$ represent the high frequency fall-off and the shape parameter, respectively. The latter determines the sharpness of the spectrum between the low-frequency constant level $M_0$ and the high-frequency fall-off with $f^{-n}$. By taking the natural logarithm of Eq. 7 we get then,



$$\ln \omega M(f) = \ln M_0 - \frac{1}{\gamma} \ln \left( 1 + \left( \frac{f}{f_c} \right)^{\gamma n} \right) \tag{8}$$

The observed source displacement spectrum data $\omega M(f)$, can be used to determine the other parameters such as $M_0, \gamma, n$ and $f_c$, in an inversion. Lastly, one of the aims of the present work can be done, coda derived moment magnitude $M_{w-coda}$ can be derived from computed Seismic moment $M_0$, using the formula given by Hanks and Kanamori (1979):

In Eq. 8 essentially an optimization problem is outlined where the obtained data source displacement spectrum data (on the left) can be modelled to estimate four unknown parameters of the source ($M_0, \gamma, n$, and $f_c$). This is accomplished through a simultaneous least-squares inversion approach. Subsequently, the moment magnitude, $M_{w-coda}$, can be computed using the modeled source parameters and seismic moment, $M_0$, employing a formula introduced by Hanks and Kanamori (1979):

$$M_{w-coda} = \frac{2}{3} \log_{10} M_0 - 6.07 \tag{9}$$

**4.2 Total Radiated Seismic Energy Estimation**

In order to estimate the radiated seismic energy first we integrate source displacement spectrum, $\omega M(f)$, and following the theoretical formula given in Gök et al. (2009). To be able to exploit the considerable part of the energy associated to the lower frequency part, observed spectrum is extrapolated to $f = 0 \ Hz$.

Here the S-wave radiated energy ($E_\beta$) can be calculated by taking integral of the energy flux in a source sphere (Patton and Walter, 1993).

$$E_\beta = \frac{4\pi}{4\rho\beta^5} \int_0^\infty |M(f)|^2 df = \frac{\pi^2 f_c^3 M_0^2}{5\rho\beta^5} \tag{10}$$

where density $\rho = 2700 \ kg/m^3$, s-wave velocity $\beta = 3.5 \ km/s$. $f_c$ and $M_0$ represent corner frequency and seismic moment estimates obtained from the inversion procedure described in Eq. 8. Here we assume that the contribution from the P-wave radiated energy ($E_\alpha$) to the total radiated energy is about 7 % of S-wave (e.g., Boatwright and Fletcher, 1984; Mayeda and Walter, 1996). Finally, the sum of P-wave and S-wave radiated energies yield total seismic radiated energy ($E_R$).

**5 Results and Interpretations**

**5.1 Coda Wave Envelope Fits**

Our preferred acoustic RTT approach to perform the forward calculation of the synthetic envelope modelling enabled the modelling of the S-wave energy propagation, thus the comparison between the synthetic and observed data, which is the portion of the seismograms directly between the S-wave arrival and the subsequent seismic coda. Previously Ryzhik et al. (1996) and Gaebler et al. (2015b) proved the validity of this approach due to the dominance of S-wave energy throughout the seismic signal, encompassing both the initial S-wave arrival and the later portions of the seismic coda. In Fig. 3, envelope fit results are presented for a selected earthquake with $M_L$ 2.9 at different frequency bands (with central frequencies of 3.0, 4.0,





6.0, 8.0, 12.0 and 16.0). The data windows length of coda wave trains ranged from -10 to 100 s relative to the onset time for all events in the present study. For the optimization process, the bounds for $g_0$ and $b$ were chosen to vary between $10^{-8} - $

$10^{-4}$ and $10^{-3} - 10^{1}$, respectively. Ultimately, unknown $g_0$, $b$, and $W$ is determined by selecting the most suitable combination of model parameters enabling the lowest error value within each frequency band. Figure 2 shows a summary of inversion process behind the envelope fitting process. According to that figure we can understand the range of the tested g0 values and further associated estimations of $b$ and $W$ at each iteration. Overall coda envelope fittings clearly illustrates that the synthetic coda envelopes are effectively required by the observed data across diverse regions within the study area and for

events with varying magnitudes. The decay of the seismic coda within time windows of up to -10 - 100 seconds is also precisely modelled, with a notable faster decay for higher frequencies. The quality of the envelope fits is comparable to those presented in previous works by Gaebler et al. (2015a), Eulenfeld and Wegler (2016), Gaebler et al. (2019), Eken (2019), and Izgi et al. (2020).

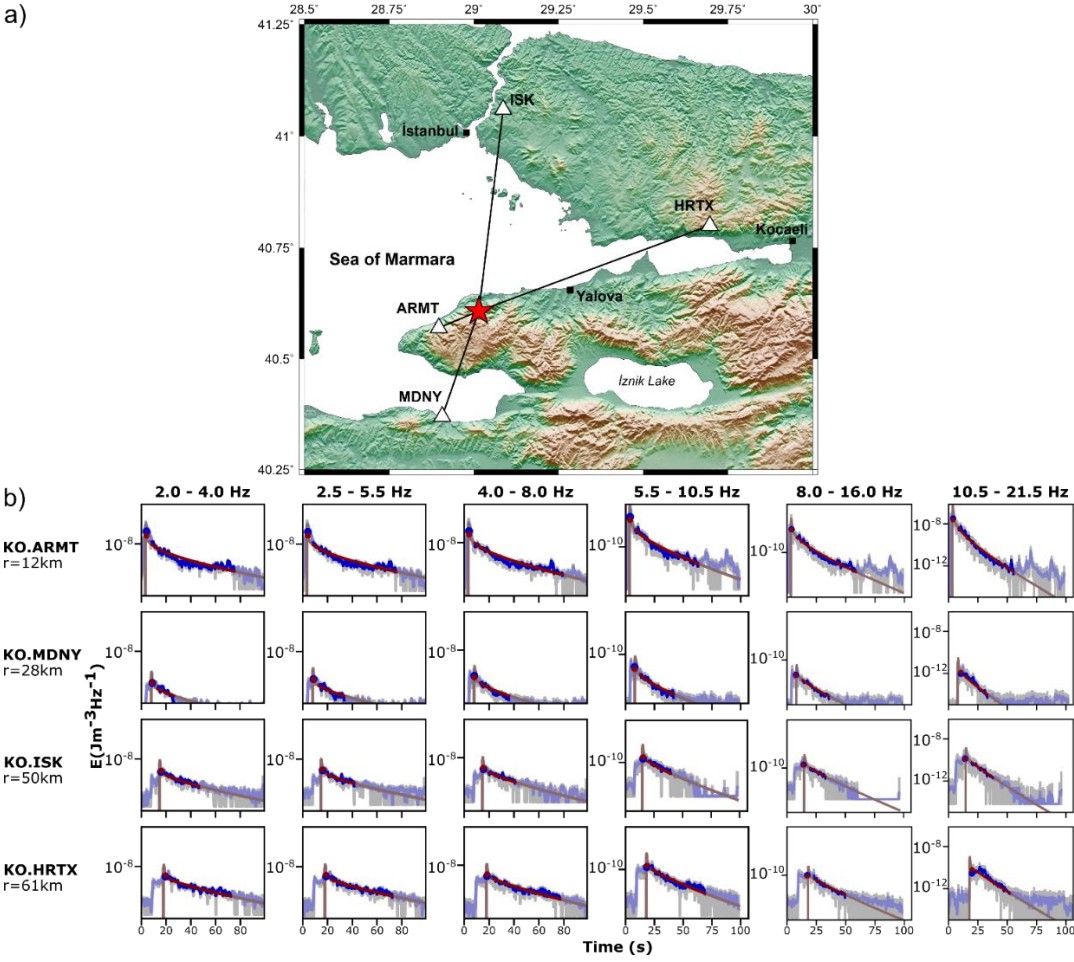

**Figure 3: a) Example event occurred in 30 November 2018 with $M_L = 2.9$ (shown with red star) and station pairs (shown with white triangles). b) Fits between observed and calculated energy densities for an example event. Grey and blue lines indicate the observed and its smoothed version, respectively. Red curves represent the computed synthetic envelopes calculated using the inversion process.**



## 5.2 Coda Wave Source Spectra

We show the observed values of source spectra from for all 303 analysed local events (compare Fig. 1) that were generated by
implementing estimated spectral source energy term $W$ at each frequency into Eq. 7. In overall, the obtained modelled spectra
(Fig. 4) appear to be well consistent with a typically expected shape of a source displacement spectrum, featuring a flat region
at around the low-frequency limit and a gradual decrease beyond a corner frequency. Earlier Walter et al. (1995) and Mayeda
et al. (2003) have shown the use of coda waves would be more advantageous in scaling-up the earthquake size as they are
rather insensitive to differences in the source radiation pattern and path effect. This mainly stems from the influence of
multiple-scattering caused by small-scale heterogeneities lead to an averaging effect on coda waves. Eulenfeld and Wegler
(2016) claimed the minor impact of radiation pattern on S-wave coda, but that it could potentially disrupt attenuation models
inferred from direct S-wave analyses if the station distribution concerning the earthquakes lacks comprehensive azimuthal
coverage. The characteristics of a source displacement spectrum, for instance, corner frequency, seismic moment, and high-
frequency falloff may be misleading in traditional approaches  (e.g., Abercrombie, 1995; Kwiatek et al., 2011) as they often
underestimate potential complexities of the source and structure by considering a fixed frequency-independent attenuation
effect described by a factor exponent $(-\pi f t Q^{-1})$ over the spectrum and an omega-square model with a constant high-
frequency falloff parameter, $n = 2$. In the present work, however, we build the source spectra based on a source term
decomposed from the effect of intrinsic and scattering attenuation. Separate estimation of source and structure-related terms
is achieved by a simultaneous inversion procedure in which the high-frequency falloff parameter changes. In line with previous
investigations (e.g., Ambeh and Fairhead, 1991; Eulenfeld and Wegler, 2016), we also noted that adopting a more realistic
methodology, as opposed to the traditional approach using the omega-square model (where $n > 3$), led to notable
discrepancies. These deviations are significant to prompt a reassessment of the widely accepted use of this model for explaining
minor earthquakes.





**Figure 4: Black squares indicate observed source displacement spectra and grey curves represents predicted source displacements spectra for all individual 303 local earthquakes.**

Previous observations (e.g., Papageorgiou and Aki, 1983; Atkinson, 1990; Joyner, 1984) indicated that source spectra, especially for large earthquakes, could be better described by models involving two corner frequencies More recently, Denolle and Shearer (2016) reported that the conventional single-corner frequency spectral model failed to explain P-wave source spectra for large thrust earthquakes ($M_w$ 5.5 and above). To overcome this, they proposed a double-corner frequency model with a lower-corner frequency associated to source duration and an upper-corner frequency indicating a shorter timescale unrelated to source duration. This upper-corner frequency also exhibits its own scaling relationship. Uchide and Imanishi (2016) reported differences from the omega-square model for smaller earthquakes following the application of a spectral ratio technique to shallow earthquakes with the magnitudes ranging between $M_w$ $3.2 - 4.0$ in Japan. They attributed these differences to fault heterogeneities, applied stress, and high-frequency falloff exponent variations. We observed high-frequency falloff parameters ($n$) ranged from $n = 0.5$ to $n = 3.5$ as they were estimated between 2 and 2.5 aligned more closely with earthquakes with $M_{w-coda} > 3.5$. The smaller magnitudes, on the other hand, exhibited a more scattered pattern in the variation of $n$ (Fig. 5). Eulenfeld and Wegler (2016) argued that a more effective strategy for inverting station





displacement spectra to estimate source parameters involves employing separate estimates of attenuation or accounting for
path effects through empirically determined Green's functions. This is, mostly, required for smaller earthquakes (with $n > 2$),
given that an omega-square model can distort estimates of corner frequency and seismic moment, particularly in the regions
of strong frequency-dependent quality factor ($Q$). Hence, we suggest, when performing inversion for source parameters, it's
crucial to incorporate independent $Q$ estimates or remove the path influence including the attenuation via empirically
determined Green's functions (Eulenfeld and Wegler, 2016).

**Figure 5: Scatter plot of $M_{w-coda}$ as a function of $M_L$ with high frequency falloff parameters $n$. Value of the $n$, is color coded with legend on the right.**

### 5.3 Coda-derived Moment Magnitude ($M_{w-coda}$)

A comparison between ML-based catalogue magnitudes (KOERI earthquake catalogues) and our $M_{w-coda}$ indicates, an
overall good accordance between them, except only for a few outliers caused by small-magnitude earthquakes. This can be
considered to be an effective usage of a straightforward model using first-order approximation for S-wave scattering with an



isotropic acoustic radiative transfer approach in relating the amplitude and decay characteristics of coda wave envelopes to the seismic moment of an earthquake at its source.

Here we introduce an empirical equation (Eq. 11) that is obtained based on a linear regression analysis between $M_{w-coda}$ and

$M_L$ magnitudes (Fig. 6). It can be used to convert local magnitudes into coda-derived moment magnitudes for local earthquakes in this region conducted a linear regression analysis between $M_{w-coda}$ and $M_L$ magnitudes (Fig. 6).



**Figure 6: Scatter plot of $M_{w-coda}$ as a function of $M_L$. Bold grey line represents the linear regression fit and dashed lines are the standard deviation.**

$M_{w-coda} = (0.6677 \mp 0.0309)M_L + 1.1914 \mp 0.09345$           (11)

In one of the earliest examples of this type of comparison, an empirical linear logarithmic relationship between seismic moments ($M_0$) and local magnitudes ($M_L$) for earthquakes near Oroville, California was established by Bakun and Lindh (1977). Other studies have explored the optimal relation between $M_w$ and $M_L$ using linear and/or nonlinear curve-fitting techniques. Instead of using a single linear fit, Malagnini and Munafò (2018) proposed two separate linear fits for $M_L$–$M_w$ data

points from earthquakes in the central and northern Apennines, Italy, divided by a crossover at $M_L = 4.3$. Various factors i.e., source scaling, crustal attenuation and/or regional attenuation, focal depth, and rigidity of the source region were considered



in the regression analyses. Relatively complicated form of empirical functions, for instance, a second-order polynomial form (e.g. Edwards and Rietbrock, 2009) associating local magnitude estimates from the Japan Meteorological Agency (JMA) with the moment magnitudes, a hybrid type of scaling relation (e.g., Goertz-Allmann et al., 2011) with a quadratic form in between $(2 \leq M_L \leq 4)$ and linear outside this range tested for Swiss earthquakes, or a quadratic form of correlation between JMA magnitudes and moment magnitudes of the seismic activity in the Fukushima Hamadori and northern Ibaraki prefecture areas of Japan (Uchide and Imanishi, 2018) have been proposed in recent years. The empirical curve derived in Uchide and Imanishi (2018) indicated a notable difference between these two magnitude scales. In their work, the graph's slope of 1/2 for microearthquakes was denoted to potential biases stemming from anelastic attenuation and presumable limitations of recording through a finite sampling interval.

Our linear empirical relation between $M_{w-coda}$ and $M_L$ magnitudes highlight an apparent move-out in Fig. 5 and Eq. 10 as being consistent with findings from early applications of the same type of coda waves modelling studies performed in different geological parts in Türkiye including central and western of the NAFZ (e.g., Gaebler et al., 2019; Izgi et al., 2020) or central Anatolia (Eken, 2019). This likely occurs due to the use of different magnitude scales for comparison. Traditional magnitude scales, such as $M_L$ based on phase amplitude measurements are prone to be affected by attenuation and path variations (Pasyanos et al., 2016). In contrast, seismic-moment-based moment magnitude ($M_w$) directly measures the strength of an earthquake from fault slip. It is derived from a mostly flat portion of source spectra at lower frequencies, making it less affected by near-surface attenuation. Relatively good agreement between coda-derived moment magnitude and local magnitude scales for the earthquakes with $M_{w-coda} > 3.5$ demonstrate the efficacy of the nonempirical method in this tectonically complicated region. This is expected for larger earthquakes whose source displacement spectra will carry more energy at lower frequencies. A similar behaviour of such coherence was observed in this region from the previous works where source characteristics of local and regional earthquakes were examined using empirical coda methods assuming simple 1-D radially symmetric path correction (e.g., Eken et al., 2004; Gök et al., 2016). Previous empirical coda envelope modelling studies (e.g., Mayeda et al., 2005b; Morasca et al., 2010) were able to estimate accurate coda-wave-derived source parameters using 2-D path-corrected station techniques that account for amplitude-distance relationships. However, noticeable outliers in our estimates (Fig. 5, 6) for the events with magnitudes less than $M_{w-coda}$ 3.5 could be attributed to potential biases in local magnitude values extracted from the catalogue as well as small biases in the intrinsic and scattering attenuation terms. Beside this such discrepancies may reflect the effects of mode conversions between body and surface waves or surface-to-surface wave scattering, which extend beyond low frequencies (Sens-Schönfelder and Wegler, 2006).

## 5.4 Self – Similarity

Accurate estimates of the seismic moment, overall radiated seismic energy of earthquakes, and associated scaled energy ($E_R/M_0$) is of great importance for clarifying dynamic modeling scenarios that are helpful to understand ground shaking for large damaging earthquakes as well as the physics behind faulting process. This is mainly because the issue of how big the



earthquake ground motions is proportional to radiated energy at the source (e.g., Brune, 1970). Whether earthquakes exhibit
self-similar scaling, or larger earthquakes differ in dynamics from smaller one has been a subject of debate for a long time.
Answering this question is essential for both making decent seismic hazard assessment and inferences on the fundamentals of
rupture dynamics during an earthquake. Over many years, it has been widely accepted that the scaled energy ($E_R/M_0$) remains
relatively fort the earthquakes of varying magnitudes from small-to-large (e.g., Aki, 1967; Kanamori and Anderson, 1975).
However, several investigations within the last two decades have observed that this ratio would tend to increase proportionally
with the seismic moment (e.g., Abercrombie, 1995; Izutani and Kanamori, 2001; Kanamori et al., 1993; Mayeda and Walter,
1996; Mori et al., 2003; Prejean and Ellsworth, 2001; Richardson and Jordan, 2002). Conversely, there exists almost equal
number of studies that advocate for a constant energy ratio (e.g., Choy and Boatwright, 1995; Ide et al., 2003; Ide and Beroza,
2001; McGarr, 1999; Prieto et al., 2004). Unfortunately, the substantial uncertainty surrounding seismic energy has led to a
diversity of interpretations of this ratio, even among researchers analysing the same dataset.
Recent advancements in scaling the size of earthquake efforts that are based on different approaches using local, regional, and
teleseismic data with different frequency contents enable to quantify scalar seismic moments, which usually exhibit small
discrepancies (more than a factor of two) for the same given event (Mayeda et al., 2005a). In contrast, the quantity of the
released seismic energy of an earthquake is rather a dynamic phenomenon and thus remains a complex endeavour, often
resulting in variations exceeding a factor of two among estimates obtained by various techniques (Pérez-Campos et al., 2003).
It requires substantial corrections that consider path and site effects across a wide range of frequencies. Further corrections for
the directivity and some other heterogeneities in source radiation pattern are equally important and must be concerned. Thus,
this ratio has been difficult and becomes the subject of recent debate among experts in the field of seismology. The uncertainty
in seismic energy calculations causes different interpretations on the apparent stress associated to the fault rigidity which may
control the energy/moment ratio or seismic energy density. To estimate seismic moment and radiated seismic energy, we
benefit from the inherent averaging characteristic of coda waves that has been earlier proved to yield notably less variability
in amplitude compared to any conventional direct phase methods (e.g., Eken et al., 2004; Mayeda et al., 2003; Shelly et al.,
2022).

The relationship between seismic moment and scaled energy observed in this study (Fig. 7) indicates that $E_R/M_0$ values
increase with the seismic moment for the crustal earthquakes with $M_{w-coda}$ 2.5 and $M_{w-coda}$ 5.7 implying these earthquakes
are likely to follow nonself-similarity. This suggests that different rupture dynamics works for large earthquakes than small
ones and the seismic energy radiates more efficiently efficient for relatively large earthquakes in the Sea of Marmara located
at the north-western part of the NAFZ. Yoo et al. (2011) previously reported that the scaled energy rapidly increases, in
particular, for smaller events ($< M_w \sim 3.3$). They attributed the size dependency of the scaled energy to the fact that the
energy radiation efficiency through seismic waves greatly varying at lower magnitudes. On the other hand, we have not
observed any distict change in the trend of scaled energy versus seismic moment almost for events in our data. We should
notice that our inference on the scaled energy is based on a first insight observation. Although the moment magnitude and
energy estimates derived from the coda in this study show very strong agreement with those reported by Mayeda and Walter



(1996) for the events of similar magnitude, a future study where we include independent waveform inversion and empirical coda modelling approaches to validate our seismic moment estimates will make our observations on self-similarity and energy
scaling more precise.

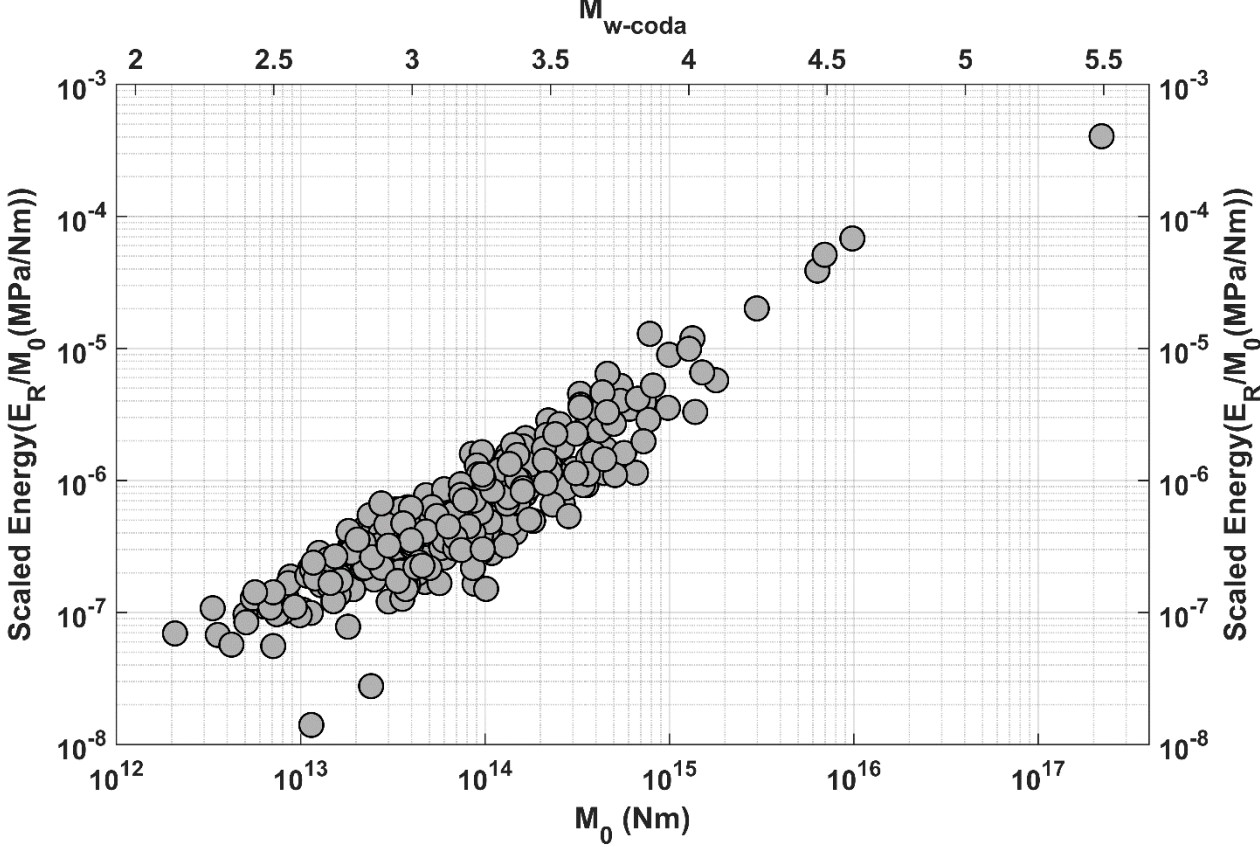

**Figure 7: Scatter plot of scaled energy ($E_R/M_0$) as a function of both $M_0$ and $M_w$.**

**6 Conclusion**

This study provides the physical measure of the released seismic energy in coda-derived moment magnitude ($M_{w-coda}$) for
minor to moderate size local earthquakes ($2.5 \leq M_w \leq 5.7$) that occurred between 2018 and 2020 in the Marmara Sea Region. This was accomplished by using digital waveform recordings taken from 49 three-component broadband seismic stations located within the study region. We used Radiative Transfer Theory for the forward calculation of synthetic coda wave envelopes during an iterative inversion procedure employing a stepwise manner to model the source properties as well as site, path effects simultaneously based on the smallest misfit between observed and synthetic envelopes. The good accordance
between $M_{w-coda}$ and $M_L$ proves the competence of this non-empirical coda wave approach to obtain reliable estimates of



source properties in this complex tectonic setting. The variability of the high-frequency fall-off parameter highlighted that for smaller earthquakes ($n > 2$), considering an omega-square model could distort estimates of corner frequency and seismic moment. This effect is particularly pronounced in regions where $Q$ (attenuation factor) exhibits strong frequency dependency. A linear regression analysis further provided an empirical relation developed between $M_{w-coda}$ and $M_L$, which can be a useful tool in the future to quickly convert catalog magnitudes into moment magnitudes for local earthquakes in the study area. Finally, the scaled energy ($E_R/M_0$) exhibits an increasing pattern with reliable coda wave-derived seismic moment estimates at almost all magnitude ranges as this implies small-to-moderate size seismic activity in the region indicates a nonself-similar scaling at their source.

**Competing Interests**

The contact author has declared that none of the authors has any competing interests.

**Acknowledgement**

This study is a part of an ongoing Ph.D. thesis by Berkan Özkan under the supervision of Assoc. Prof. Dr. Tuna Eken. Digital waveform recordings of local earthquakes analysed in the present study and station metadata were acquired from the data management facilities of the Kandilli Observatory and Earthquake Research Institute (KOERI). Berkan Özkan, BÖ, Tuna Eken, TE, and Tuncay Taymaz, TT, would like to thank Istanbul Technical University, the National Scientific and Technological Research Council of Türkiye (TÜBİTAK), Turkish Academy of Sciences (TÜBA) in the framework for Young Scientist Award Program (TÜBA-GEBİP), the Science Academy Chamber–Türkiye (BAGEP), and the Alexander von Humboldt Foundation for further providing computing facilities and other relevant computational resources through Humboldt-Stiftung Follow-Up Programme. A python package for Qopen utility, which implements the proposed inversion scheme used in the present study was made available by Tom Eulenfeld at https://github.com/trichter/qopen.

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
