# Peer review of "Coda-derived source properties estimated using local earthquakes in the Sea of Marmara, Türkiye"

_EGUsphere, 2024_

## Author Response (AR1)

**Reply Letter**

**Reviewer 1:**

This study makes a significant contribution to the field of seismology, particularly through its innovative use of coda wave modeling to estimate moment magnitudes of earthquakes in the Sea of Marmara. The comprehensive dataset, consisting of 303 local earthquakes recorded at 49 seismic stations between 2018 and 2020, enhances the robustness of the findings. The inversion of coda wave traces across twelve frequency ranges is a noteworthy methodological strength, providing detailed insights into the source displacement spectrum. The alignment between coda-derived moment magnitudes and standard local magnitude estimates is promising, despite some discrepancies for smaller earthquakes. The introduction of an empirical relationship between local magnitude and coda-derived moment magnitude is particularly valuable for future applications.

**Reply**: We appreciate for your overall description and constructive opinion regarding the present work, and for your further comments and suggestions, which were all very encouraging and valuable for us in improving the manuscript. Below we address all these points with relevant clarifications, corrections in the text, and further implementations on the figures.

Some minor comments:

Abstract

Line 19: What the authors mean by "devising an emprical relationship". Do they suggest a new methodology or do they just mean observed by the comparison.

**Reply:** We acknowledge the concerns raised by the reviewer and wish to clarify that our methodology does not entail the development of an entirely novel approach. Instead, we have employed a regression analysis to derive a novel empirical relationship between conventional local magnitude and coda-derived moment magnitude estimates. Consequently, we have modified the Abstract to replace the term "devised" with "obtained".

Introduction

Line 21: The authors may consider to cut the sentence into two or avoid using the phrase of "such moment magnitude estimates".

**Reply:** As you suggested, we have removed the sentence in the comment in order to preserve the formal sentence structure, and highlighted examples of source parameters in parentheses.

Line 36: Could the authors discuss this sentence in more detail. "It is widely recognized that local or regional coda waves mainly consist of scattered waves." The shape of the coda wave is mainly driven by other attenuation types. Thus, this statement can be discussed further or do they refer to attenuation in general instead of scattering?

**Reply:** We agree with the reviewer. The shape of coda waves is governed by a combination of intrinsic and scattering attenuation mechanisms. However, our objective in this statement is to highlight that coda waves are solely composed of scattered waves, as a consequence of the interaction between seismic waves and scatterers within the volume located between the earthquake source at the foci and the station. These scatterers reflect small-scale inhomogeneities within the crust.

Data

The authors may explain why the central frequencies chosen as these.

**Reply:** The objective of the present study was to obtain coda-derived moment magnitudes for local events with the smallest possible magnitude local earthquakes. Given that the majority of earthquakes in our database are relatively small and possess a high-frequency energy content, we endeavoured to model the high-frequency portion with a maximum center frequency of 16 Hz. This was done with the intention of maintaining consistency with the coda wave energy portion at high frequency, as determined through our spectrogram analyses (please see Fig. R1 and R2). It should be noted that implementing higher center frequencies than 16 Hz did not result in any drastic change to the source displacement spectra of the example events. Furthermore, the objective was to model coda waves for the lowest possible frequency, particularly in the context of relatively large earthquakes with potentially low-frequency contents in their source displacement spectra. In the present study, we were able to utilize a minimum frequency content of 0.3 Hz on the samples. Frequency bands were created at equal intervals with an octave difference between the upper and lower corner frequencies, thereby enabling the most efficient modelling of seismic moment, corner frequency and high-frequency fall-off parameters.

[Figure]

Figure R1. Example waveforms of the vertical component and corresponding spectrograms observed at station KRBG for 26 September 2019 ML=5.7 (top-left), 24 September 2020 ML=3.7 (top-right) and 5 January 2019 ML=4.1 (bottom) events.

[Figure]

Figure R2. Example waveform of the vertical component and corresponding spectrograms observed at station ADVT for 26 September 2019 ML=5.7 (top-left), 24 September 2020 ML=3.7 (top-right) and 5 January 2019 ML=4.1 (bottom) events.

It would be nice to see a waveform example with chosen parameters shown. When the datasets of similar studies considered, the dataset seems extremely lucky.

**Reply:** We thank the reviewer for this comment. Actually, in Fig. 3 we fulfil this wish somewhat by presenting observed coda envelope examples calculated using some selected station recordings of a sample event by performing a simple Hilbert transform.

The authors could explain why they chose minimum of 4 stations for each event.

**Reply:** As the Marmara Sea region is large for this type of study, we could not use a specific list of stations for each event. For the solutions with a small number of stations (e.g., <4), our coda-based source property estimates showed significant discrepancies from those published in the catalogue. During our analyses, we found that we were able to obtain consistent solutions from 303 events with coda wave records observed at least 4 stations. Furthermore, this number as a minimum is often acceptable and used in different fields of seismology, for example in simple localization procedures.

How many earthquakes are presented in Figure 1. I believe the authors are showing 375 events. Visually they seem a lot less probably because of the circle size. At least mentioning the earthquake count in the figure caption would help.

**Reply:** Thank you for this comment. Figure 1 shows only locations of 303 earthquakes for which source parameters can be calculated. We have modified the figure caption accordingly based on your suggestion.

Method

Why the Figure 2 shows 5.5 to 10.5 Hz whereas in data section the authors presented different central frequencies.

**Reply:** In the data section the central frequencies of the frequency bands are given (0.3, 0.5, 0.7, 1, 1.4, 2.0, 2.8, 4.0, 6.0, 8.0, 12.0, 16.0 Hz), not the frequency bands. The frequency bands were selected in an octave width starting from 0.3 Hz up to 16 Hz center frequencies with a step size of 0.5 Hz. In Figure 2, the frequency band 5.50-10.50 Hz is chosen as an example to illustrate the optimization process and has a central frequency of 8 Hz.

Also at Figure 2, I do not see any colors as the authors mentioned in the figure caption. I am not sure if it's caused by the printing but I can see the colors on the other figures but not this one. Although one might figure out where blue cross is, it would be nicer to show it.

**Reply:** Yes, unfortunately the figure was accidentally rendered in greyscale. The version with a blue cross has been added.

Result and Interpretations

Line 266: In which way the result fits are comparable to the other mentioned studies.

**Reply:** We meant that our resulting coda fits are comparable in the sense that the overall misfit during the optimisation process of the observed coda envelope showed similarities to those of previous misfit values (Fig. 2 of the manuscript) obtained from the same approach performed on the north-western and central NAFZ (Izgi et al., 2020; Gaebler et al., 2019) and central Anatolia (Eken, 2019).

Line 274: Authors state that Figure 1 shows 303 selected events. Is this information correct? Please check also my comment related to Figure 1 under Data section.

**Reply:** Yes, it is correct. We have carefully checked the GMT script and our input files used to generate this figure. As a result, we can confirm that we have plotted the locations of all 303 events for which source parameters could be calculated in this study.

Line 298: "two corner frequencies More recently," dot is missing between two sentences.

**Reply:** Thank you notifying for this mistake. The missing dot has been added to the revised version.

Line 319: "A comparison between ML-based catalogue magnitudes (KOERI earthquake catalogues) and our $Mw-coda$ indicates, an overall good accordance between them, except only for a few outliers caused by small-magnitude earthquakes." Can the authors provide a percentage of "good accordance" and how many outliers there are and would it be possible to somehow mark them on figures 5 and 6.

**Reply:** An overall correlation between two magnitude scales is 0.78. I guess this should be enough to give an idea about the goodness of accordance in a statistical manner.

Line 368: There seems like to be a font issue. Also, the sentence is hard to understand.

**Reply:** Yes, font size was different for few words, we have changed it to 10 pt. in the revised manuscript to maintain the consistency.

The discussion highlights the good agreement between coda-derived moment magnitudes (Mw-coda) and local magnitudes (ML) for earthquakes with Mw-coda > 3.5, while noting potential biases for smaller events due to anelastic attenuation and recording limitations. Could you elaborate on how these biases specifically impact the empirical relation between Mw-coda and ML, and what methodologies could be adopted in future studies to better account for or mitigate these biases, particularly for smaller magnitude earthquakes?

**Reply:** The consideration of unrealistic attenuation and the inadequate representation of the high-frequency content of local events due to the limitations of the recording capabilities of permanent or temporary seismic networks as their size decreases are the problems in the recovery of frequency-dependent true source terms in coda wave modelling. The methodology used here essentially aims to deal with the former by modelling it within the same stepwise inversion scheme. Furthermore, in the representation study we compare our *Mw-coda* estimates with the corresponding local magnitudes of the same earthquakes published by KOERI. Apparent outliers, especially for smaller events (Mw<3.5), are likely to occur because *ML* based on phase amplitude measurements are susceptible to attenuation and path variations. In order to better understand the potential impact of these artefacts, our future goal is to perform a comparison between our Mw-coda and another moment magnitude estimate that can be obtained by independent waveform inversion.

Conclusion

Line 419: "A linear regression analysis further provided an empirical relation developed between $Mw-coda$ and $M_L$ , which can be a useful tool in the future to quickly convert catalog magnitudes into moment magnitudes for local earthquakes in the study area." Is the process really quick for conversion compared to other methods? If so that would be nice to see in discussion part. Also I presume this analysis would be useful in any study area for local earthquakes so that the authors maybe rephrase the sentence not emphasizing on 'the' study area.

**Reply:** Well, we meant that by inserting the necessary parameters into this new empirical relation found for the region, one can easily perform a conversion from *ML* to *Mw-coda* and vice versa. Other coda calibration methods (e.g., Mayeda et al., 2003) can be used to estimate *Mw-coda* values for the earthquakes in our database, depending on any magnitude scale originally used to identify the size of these earthquakes and further validate the coda method. Regression analysis can also provide another empirical relationship. To avoid any misunderstanding, we have removed the word "fast" from the statement.

On the other hand, different segments of the NAFZ from east to west may have different crustal heterogeneities affecting the intensity and depth of these scatterers. Therefore, we would still like to keep "the study area" in the sentence, as the empirical relationship found in this study cannot be considered unique for the entire NAFZ in this complex tectonic setting.

Overall, this study is methodologically rigorous and offers meaningful advancements in understanding earthquake dynamics and seismic hazard assessment.

**Reply:** Once again, we would like to thank the reviewer for these very constructive and encouraging thoughts on our work.

**Reviewer 2:**

In this paper, Özkan et al., estimate the source parameters of 303 ML [2.5 – 5.7] earthquakes in the Sea of Marmara region, NW Türkiye, using a coda-wave modelling approach. The authors derived a scaling relationship between ML and MW coda. One main result is the reporting of non-self-similar earthquake behaviour based on in the increasing ratio of scaled energy (E0/M0) with seismic moment. The topic is relevant to the community, and, especially to the Marmara region, there are not many studies on this topic, making this contribution particularly valuable. The manuscript is well written and (to some extent) logically organized.

**Reply:** We appreciate for your overall description and constructive opinion of the present work. Your further comments and suggestions, which appear to be the result of a careful reading, have helped us considerably to improve the quality and content of the manuscript. In the following, we address all these points with relevant clarifications, corrections in the text and further implementations in the figures.

Some recommendations are provided below:

I feel certain imbalance in the manuscript between the introduction to the region and the methodology, which is extensive and detailed, and the results, interpretation and conclusions, which I feel are rather scarce and limited. For example, in Section 2, the description focuses on the potential for large earthquakes from different fault segments. However, this important issue is never brought up in the results, or, more importantly, on the interpretation of the results. These two parts of the paper seem somewhat disconnected. The estimation of the Coda parameters was quite extensive and difficult. What new information do these results provide, in the context of the Marmara region and the ongoing deformation there? I would suggest working on extending this part.

**Reply:** We are very grateful to the reviewer for raising this issue. The main output of this work is *Mw-cod*a, which represents a direct physical quantity of the energy released at the earthquake foci. Reliable estimates of the seismic moment and relevant earthquake magnitudes are of great importance for further seismic evaluations in earthquake-prone regions, as they can provide useful insights into the length and depth of the ruptured fault segment and the average slip during the co-seismic period. However, it is worth mentioning that our main purpose in the present study is to only demonstrate the usefulness of the coda-wave modelling approach for the regions with different tectonic settings. In short, we have successfully applied the method to the events in the study area, which are not selected considering any spatio-temporal pattern in the region, established an empirical relationship between conventional and new coda-derived magnitude scale, and further reported a possible breakdown of the self-similarity hypothesis. In this regard, our paper has nothing too much to discuss the potential for large earthquakes from different fault segments, even though its original version presented an unnecessary and redundant amount of literature examples investigating the seismogenic character of the NAFZ, focusing mainly on the Marmara region. Thus, we have essentially summarised this part in such a way that it only gives the main reason why having proper magnitude estimates for the region is meaningful.

Unfortunately, I have strong concerns about the reported breakdown in the self-similarity of the earthquakes for this region. As the authors note, many artefacts can lead to such a plot (Figure 7). The fit looks so sharp that it is hard to believe that this is a real feature. Simply reporting this as a "result that deserves further analysis in the future" may be misleading to the community, who may believe that these are rigorous results. Instead, as this is a major result of their analysis, I encourage the authors to evaluate more deeply the various artifacts that can promote such an artificial trend. Some of them include (1) limited bandwidth, (2) Biases in the station See Cocco et al., (2016) for a broader overview on factors that could potentially contribute to biases in this relation.

**Reply:** We fully understand the reviewer's concerns on this issue. In response to the reviewer's comment, a detailed discussion concerning potential artefacts has been included in Section 5.4. The emphasise, on the other hand, placed on the end of Section 5.4 for the reference to a future study in the last sentence of the Results and Interpretations section was initially indented to show our future plans for investigating the same phenomenon using a completely independent method on this or another data set. However, we have since removed that statement to avoid any confusion.

Introduction: there is a certain amount of repetition, which can also be seen in the repetition of sequences of references. For example, see lines 110 and 140.

**Reply:** We are very grateful to the reviewer. We have realized this issue, in particular, for the Section 2. While we summarize this section in accordance with the reviewer#2's first comment above, we have removed those redundant and repeating several repeating statements we have also realized several repeating statements.

Section 2: The selected data set extends beyond the Main Marmara Fault and also covers the Armutlu Peninsula. I suggest to also add a description of the seismological and tectonic processes of this region, as it seems to be an important part of your data set (see e.g., the event shown in Figure 3). Some literature from the region includes Martínez-Garzón et al., (2019; 2021); Bocchini et al., (2022).

**Reply:** This would be nice, but in the revised version of the manuscript we have made a substantial summary to avoid unnecessary literature on the seismogenic characteristics of the study region. However, we thank the reviewer for suggesting these literature examples. In particular, we found Martínez-Garzón et al. (2019; 2021) relevant and useful to better highlight the issue of seismic hazard potential in the region, so we have implemented it in the revised version of Section 2.

Section 3: Why there is a need to restitute if these are broadband stations? Also, is it necessary to give only central frequencies instead of detailing both frequency cutoffs? It would help to exclude bandwidth limitation problems.

**Reply:** Yes, they are broadband stations. But since our coda wave modelling approach used here does not depend on a differential strategy such as taking spectral ratio and further the digital recordings of the earthquakes utilized in this work were acquired from the instruments with different sensor types, we have restitute in order to better mimic the actual ground motion on seismograms.

Other comments:

L 164: "The noise level before P-wave onset was removed" How?

**Reply:** We apologise for any confusion. What we actually meant to say was that we disregarded the pre-P wave noise level. We have changed this statement in the revised text by replacing "removed" with "disregarded".

L 165: Why these choices on window length, and do small variations play a role ?

**Reply:** We decided on these parameters, which gave the best coda envelope fits, after a series of trial-and-error runs on several example events. During these tests we found that a few seconds of variation did not drastically change the overall magnitude estimates.

L 168: SNR from where?

**Reply:** The SNR is simply calculated using the noise and signal power prior and after the P-wave onset.

L 172: "Earthquakes" should be "Stations" ?

**Reply:** No, we really intended to talk about earthquakes because the phrase describes the criteria used to eliminate events in our automated processes.

L 185: parameter (no s in the end).

**Reply:** Corrected.

L 201: THE three main steps… […] is given à are given

**Reply:** Corrected.

L 203: parameter (no s in the end).

**Reply:** Corrected.

Figure 2 caption: Description of the y axis of the small insets and the actual y label of the insets do not match. Which one is correct?

**Reply:** The label in the figure ("ln(E_obs/(GiRi))") is the correct one. We have corrected the figure captions accordingly.

L 274: Sentence weird. Please revise.

**Reply:** Thank you. We have reformulated the sentence.

Figure 4, y label: please add units.

**Reply:** We appreciate for noticing this. The unit of the y-axis is Nm. We have corrected it in the revised manuscript.

L 370: smaller oneS.

**Reply:** Corrected.

L 373: relatively "fort"?

**Reply:** We corrected the statement by adding "nearly constant".

L 396: efficient is doubled.

**Reply:** Corrected.